# Towards Accurate and Fair Cognitive Diagnosis via Monotonic Data Augmentation

**Zheng Zhang**[1,2]    **Wei Song**[1,2]    **Qi Liu**[1,2]    **Qingyang Mao**[1,2]    **Yiyan Wang**[3]
**Weibo Gao**[1,2]    **Zhenya Huang**[1,2]    **Shijin Wang**[2*]    **Enhong Chen**[1,2]

1: University of Science and Technology of China
2: State Key Laboratory of Cognitive Intelligence
3: Beijing Normal University
{zhangzheng,sw2,maoqy0503,weibogao}@mail.ustc.edu.cn;
{qiliuql,huangzhy,cheneh}@ustc.edu.cn;
wangyiyan@mail.bnu.edu.cn;
sjwang3@ifytek.com

## Abstract

Intelligent education stands as a prominent application of machine learning. Within this domain, cognitive diagnosis (CD) is a key research focus that aims to diagnose students' proficiency levels in specific knowledge concepts. As a crucial task within the field of education, cognitive diagnosis encompasses two fundamental requirements: accuracy and fairness. Existing studies have achieved significant success by primarily utilizing observed historical logs of student-exercise interactions. However, real-world scenarios often present a challenge, where a substantial number of students engage with a limited number of exercises. This data sparsity issue can lead to both **inaccurate** and **unfair** diagnoses. To this end, we introduce a monotonic data augmentation framework, CMCD, to tackle the data sparsity issue and thereby achieve accurate and fair CD results. Specifically, CMCD integrates the monotonicity assumption, a fundamental educational principle in CD, to establish two constraints for data augmentation. These constraints are general and can be applied to the majority of CD backbones. Furthermore, we provide theoretical analysis to guarantee the accuracy and convergence speed of CMCD. Finally, extensive experiments on real-world datasets showcase the efficacy of our framework in addressing the data sparsity issue with accurate and fair CD results.

## 1  Introduction

Intelligent education is a significant domain within machine learning that focuses on exploring students' learning patterns. In the past decades, research in this interdisciplinary field has garnered substantial attention from scholars across various disciplines [29, 26, 10, 45], including education, machine learning, and psychology. Within intelligent education, cognitive diagnosis (CD) stands out as a crucial research area that aims to measure students' proficiency levels in specific knowledge domains, such as Geometry [39, 15, 7, 21]. For instance, as illustrated in Figure 1, students practice some exercises $e_1, e_2, e_3$ and obtain associated responses indicating correctness, which are then utilized to perform CD to infer the students' mastery levels of the corresponding concepts. With a comprehensive understanding of students' abilities, CD can be applied to various applications, including student assessment [4, 56] and educational recommendation systems [19].

As a crucial task within the field of education, CD encompasses two fundamental requirements: 1) **Accurate Diagnoses**: It is imperative to precisely evaluate students' mastery of knowledge to facilitate

---

*Corresponding Author

38th Conference on Neural Information Processing Systems (NeurIPS 2024).

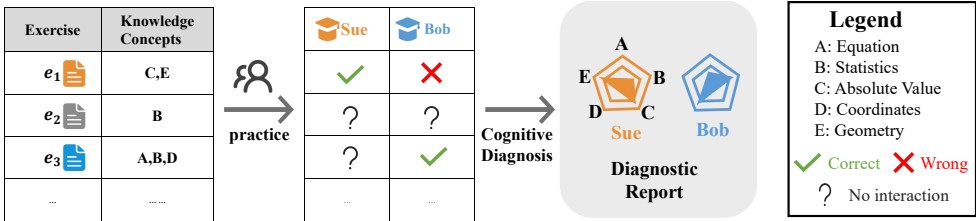

Figure 1: An illustrative example of cognitive diagnosis.

various downstream applications. For instance, this accuracy enables teachers to deliver personalized instruction, tailored to the specific needs of each individual student. 2) **Fair Diagnoses**: CD plays a pivotal role in high-stakes examinations like the GRE [30], significantly influencing individuals' developmental opportunities. Therefore, ensuring fairness in CD is of paramount importance. In this context, fairness means different student groups divided by sensitive attributes (e.g., gender, race) should be treated fairly and equally.

To accomplish these dual objectives, prior research has made notable strides by harnessing historical logs that document students' interactions with exercises. However, in practical scenarios, many students can only interact with a limited number of exercises within the vast exercise pool, leading to the *data sparsity* issue that may cause both **inaccurate** and **unfair** diagnoses. A detailed analysis is available in Section 4. Currently, several approaches have been proposed to mitigate this issue from a model-centric perspective, which focus on developing more complex architectures to address the challenges posed by data scarcity [14, 7]. However, the incorporation of additional architectures often compromises the model's interpretability, rendering it unsuitable for high-impact educational environments. For instance, in high-stakes exams like the GRE, only IRT [29], a classical CD model, has been applied due to their statistical superiority [30]. While other model-based models, despite mitigating data sparsity to some extent, remain unused because of their lack of interpretability.

In contrast to these model-based approaches, in this paper, we tackle the data sparsity issue from the perspective of data augmentation without altering the model architecture. However, we have encountered the following challenges: *1) Monotonicity Assumption.* The Monotonicity Assumption is a fundamental theoretical premise in the field of CD. Specifically, it posits that a student's proficiency exhibits a monotonic relationship with the probability of providing a correct response to an exercise. Using Figure 1 as an example, Sue, who correctly responds to exercise $e_1$, is considered to possess a higher proficiency level in the related concept C (i.e., Absolute Value) compared to Bob, who provides an incorrect response. This ensures the interpretability of CD, contributing to its widespread acceptance and application. Therefore, maintaining the model's monotonicity assumption during the data augmentation process is crucial. *2) Theoretical Guarantees.* As data augmentation emerges as a new paradigm in the CD domain, it becomes imperative to establish corresponding theoretical guarantees for this novel approach.

To address these challenges, we propose a monotonic data augmentation CD framework, CMCD, complemented by theoretical guarantees. Specifically, we integrate two data augmentation constraints confronting the monotonicity assumption. For each student, we generate fake students by reversing one of his responses while keeping other records unchanged to train CD models. As illustrated in Figure 2, Bob's wrong answer to $e_1$ is reversed to generate C-1 while his right answer to $e_6$ is reversed to generate C-2. Following the monotonicity assumption, we assume that Bob's proficiency the corresponding knowledge concept is lower than that of C-1 yet higher than that of C-2. Moreover, we provide theoretical analysis to guarantee CMCD's advantage in **accuracy** and **convergence speed**. Finally, we conduct extensive experiments on real-world datasets, demonstrating the effectiveness of our method across various CD models. Our key contributions can be summarized as follows:

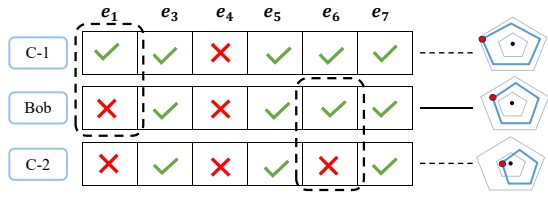

Figure 2: An example of monotonic augmentation.

- **Problem Emphasis.** Our paper strongly emphasizes the connection between data sparsity and inaccurate and unfair diagnoses. To the best of our knowledge, we are the first to highlight the relationship between data sparsity and educational fairness.

- **Framework Design.** From the perspective of data augmentation, we combine the monotonicity assumption in CD to propose a general framework, CMCD, which comes with theoretical guarantees of accuracy and convergence speed.
- **Experimental Evaluations.** Extensive experiments on real-world datasets demonstrate the effectiveness of our approach.

## 2 Related Works

### 2.1 Cognitive Diagnosis

CD plays a crucial role in various real-world educational scenarios [48, 49], including student assessment [36, 31] and educational recommender systems [24, 19]. Initially rooted in psychometrics, CD models such as Item Response Theory (IRT) [29] have been widely used, particularly in the context of GRE [30]. Later, Multidimensional Item Response Theory (MIRT) [35] was introduced to extend the single-trait features in IRT to multidimensions. While these models were effective and provided interpretable diagnostic results based on psychometric theories, they heavily relied on handcrafted interaction functions and could only leverage users' numerical response records, which were often affected by data sparsity [25]. With the advancements in machine learning, researchers began developing CD models from a machine-learning perspective. One notable model in this regard is NCDM [39], which employed neural networks to learn the interactions between students and exercises, yielding satisfactory results. However, the majority of existing works have primarily focused on designing more complex architectures to alleviate data sparsity. In contrast to traditional model-based approaches, this paper embraces a data-centric perspective and introduces CMCD, which comes with strong theoretical guarantees and is adaptable for integration with all cognitive diagnosis models.

### 2.2 Data Augmentation

Data augmentation is a classical technique to address the problem of insufficient training data in the machine learning community [42]. Over the past few years, this concept has achieved significant success in the fields of neural language processing (NLP) [13, 57, 32] and computer vision (CV) [38, 2, 6]. Later, this idea was applied to user modeling tasks, which necessitated the simulation of user-item interaction data. The work most similar to ours involves data augmentation of user-item interactions in recommendation systems. They typically answer questions such as: "What would happen if...?" Through these responses, they can generate additional virtual data, thereby alleviating the issue of data sparsity. For instance, Wang et al. [43] addressed the issue of data sparsity in the sequential recommendation and provided an answer to the question: "What would a user like to buy if their previously purchased items had been different?". Similarly, Xiong et al. [44] focused on review-based recommendation and answered the question: "What would be the user's decision if their feature-level preference had been different?" Although these works have shown promising results, when applied to cognitive diagnosis, they overlook the monotonicity assumption, a classic theory in education. Additionally, they lack corresponding theoretical guarantees. In this paper, we aim to bridge these gaps to alleviate data sparsity in cognitive diagnosis more effectively.

## 3 Preliminaries

### 3.1 Cognitive Diagnosis and Models

In this subsection, we formally define the CD problem. Assume there are $m$ students, $n$ exercises, which are denoted as $S = \{s_i\}_{i=1}^m, E = \{e_j\}_{j=1}^n$. Assume each student $s_i$ has records on exercises indexed by $Q_i \subset \{1, 2, \ldots, n\}$, the response logs $R_i$ of student $s_i$ are a set of triplets $(s_i, e_j, y_{i,j})$, where $j \in Q_i$, $y_{i,j} \in \{0, 1\}$ is the score obtained by student $s_i$ on exercise $e_j$. Given response logs $R_i$ of student $s_i$, the goal of CD is to mine the proficiency $\theta_i$.

Cognitive Diagnosis Models (CDMs) are developed to depict student's proficiency level on specific knowledge concepts based on her responses to several test items. To do this, an objective function is used to train CDMs on the student performance prediction task. More concretely, CDMs are expected to minimize the difference of the predicted probability $p_j(\theta_i)$ of a student $s_i$ giving the right response

to the exercise $e_j$ between the true response $y_{i,j} = p_j(\theta_i)$. In this paper, following the classical works [14], we adopt the cross-entropy loss, the goal is,

$$\mathcal{L} = \frac{1}{m} \sum_{i=1}^{m} l_i(\theta_i) = -\frac{1}{m} \sum_{i=1}^{m} \sum_{j \in Q_i} [y_{i,j} \log p_j(\theta_i) + (1 - y_{i,j}) \log(1 - p_j(\theta_i))]. \tag{1}$$

In the past decades, lots of CDMs have been proposed such as IRT [29, 11], MIRT. Generally, CDMs contain two parts: (1) the representations of trait features and (2) the interaction function. For example, IRT models each student $s_i$ as a proficiency variable $\theta_i$, each exercise as a discriminating factor $\alpha_j$ and a difficulty factor $\beta_j$, and a logistic function is used to forecast the likelihood that student $s_i$ will answer exercise $e_j$ correctly based on a logistic function [2], i.e., $p_j(\theta_i) = 1/(1 + e^{\alpha_j(\theta_i - \beta_j)})$.

## 3.2 Fairness in Cognitive Diagnosis

With the advancement of machine learning technologies [20, 9], such as large language models [55], which have found widespread applications in many important scenarios [51, 52], trustworthy AI has become a very important topic [18, 27, 28, 34, 33, 17]. Among these concerns, the issue of fairness has garnered widespread attention [22, 37, 8, 41, 53, 47, 40]. Given that cognitive diagnosis holds a fundamental position in the field of education and is extensively applied in high-stakes exams such as the GRE [30], which significantly shapes individuals' developmental opportunities, ensuring fairness in cognitive diagnosis becomes of utmost importance. In this paper, we follow the widely accepted group fairness definition proposed by Li et al. [23], which states that a fair model should provide the same level of utility performance for different user groups.

For analytical convenience, we focus on the case where the sensitive attribute is binary, which can be easily extended to multiple values. The student group can be divided into two groups based on their sensitive attributes, denoted as $G_0$ and $G_1$, where $U = G_0 \cup G_1$, $G_0 \cap G_1 = \varnothing$. We denote the number of samples in each group as $m_0$ and $m_1$. Inspired by previous works [54, 50, 23], in this paper, fairness in cognitive diagnosis is defined as follows, where a lower value indicates better fairness performance:

**Definition 3.1** (Fairness in Cognitive Diagnosis)**.**

$$GF = \left| \frac{1}{m_0} \sum_{s_i \in G_0} \mathcal{M}(s_i) - \frac{1}{m_1} \sum_{s_i \in G_1} \mathcal{M}(s_i) \right|, \tag{2}$$

where $\mathcal{M}$ represents a metric for evaluating utility performance, such as MAE or MSE score, and $\mathcal{M}(s_i)$ denotes the utility performance for student $s_i$.

After introducing the fairness definition in cognitive diagnosis, our objective extends beyond accurately identifying students' proficiency. We should also strive to meet fairness requirements, with the goal of minimizing $GF$.

# 4 Data Sparsity Analysis

In this section, we explore data sparsity in cognitive diagnosis through the real-world dataset, Math (detailed information will be introduced in the Experiment). Firstly, we conduct an analysis of data sparsity within the dataset, partitioning it based on the number of responses per student. The statistical data is illustrated in Figure 3(a). From the figure, it is evident that the majority of users have a low log count, highlighting the widespread presence of data sparsity. Following the methodology outlined in [46], we classify records with fewer than 50 responses as the sparse group. Next, we validate whether data sparsity affects the accuracy and fairness of cognitive diagnosis.

**Inaccurate Result**    From the perspective of accuracy, we conduct a comparative analysis between the sparse and non-sparse groups, as illustrated in Figure 3(b). The results indicate a notable decrease in performance for the sparse group across different backbones compared to the non-sparse group. This outcome strongly implies that data sparsity can result in inaccurate diagnostic results.

---

[2]Here we adopt two-parameter logistic IRT model.

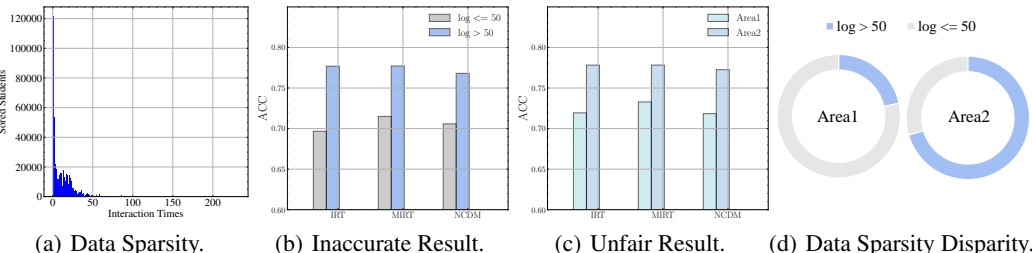

| (a) Data Sparsity. | (b) Inaccurate Result. | (c) Unfair Result. | (d) Data Sparsity Disparity. |

Figure 3: The impact of data sparsity on CD models

**Unfair Result** From the perspective of fairness, we examine the performance of the two groups divided by sensitive attributes. In the Math dataset, the student group is divided into Area1 and Area2 by sensitive attribute region, with Area1 representing the disadvantaged group. The results are depicted in Figure 3(c), unveiling significant variations in performance outcomes among different groups and indicating the presence of unfairness. Following this, we delve into the disparity in sparsity levels between the two groups, as shown in Figure 3(d). The results highlight significantly divergent sparsity levels, notably with the disadvantaged groups exhibiting higher sparsity. This discrepancy is attributed to the likelihood that disadvantaged groups may have fewer opportunities to engage in relevant online learning platforms compared to their advantaged counterparts. This statistical outcome directly implies that data sparsity has a greater impact on the disadvantaged group (Area1). Based on this observation, we can conclusively state that data sparsity can lead to unfair diagnostic results.

So far, we have validated that data sparsity in cognitive diagnosis can result in inaccurate and unfair diagnostic outcomes. Consequently, mitigating the issue of data sparsity in cognitive diagnosis is of paramount importance. In the following sections, we will demonstrate how to address data sparsity from the perspective of data augmentation.

## 5 The Proposed Framework

In this section, we propose a monotonic data augmentation framework, CMCD, as depicted in Figure 4. It can address the issue of data sparsity and ensure accurate and fair diagnostic results. Firstly, we present the constraints of our proposed monotonic data augmentation approach. Subsequently, we offer theoretical assurances regarding the effectiveness of CMCD.

### 5.1 Monotonic Data Augmentation

In CD, there is a fundamental assumption called the monotonicity assumption [1], which plays a vital role in ensuring interpretability. In this

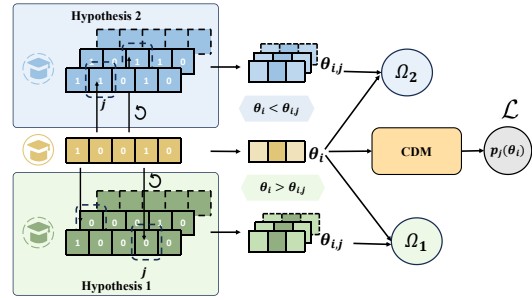

Figure 4: The CMCD framework.

paper, we introduce a novel approach that integrates the monotonicity assumption with data augmentation. This combined technique allows us to generate more realistic student response data that conforms to educational principles. As a result, we can effectively address the challenge of data sparsity. The monotonicity assumption is defined as follows:

**Monotonicity Assumption** The probability of correct response to the exercise is monotonically increasing at any dimension of the student's knowledge proficiency.

Building upon the monotonicity assumption, suppose two students $s_i$ and $s_{i'}$ have answered exactly the same exercises, i.e., $Q_i = Q_{i'} \triangleq Q$, we propose two specific hypotheses as follows:

**Hypothesis 1**: If $s_i$ answered exercise $e_j$ correctly, while $s_{i'}$ answered $e_j$ incorrectly, and answers for other exercises are the same, then the proficiency level of $s_i$ should not be lower than $s_{i'}$. Formally, for only one $j \in Q, r_{i,j} = 1, r_{i',j} = 0$, while for $k \in Q, k \neq j, r_{i,k} = r_{i',k}$, we have $\theta_i \geq \theta_{i'}$.

**Hypothesis 2**: If $s_i$ answered exercise $e_j$ incorrectly, while $s_{i'}$ answered $e_j$ correctly, and answers for other exercises are the same, then the proficiency level of $s_i$ should not be greater than $s_{i'}$. Formally, for only one $j \in Q, r_{i,j} = 0, r_{i',j} = 1$, while for $k \in Q, k \neq j, r_{i,k} = r_{i',k}$, we have $\theta_i \leq \theta_{i'}$.

To leverage these hypotheses and enhance the training data to mitigate the data sparsity issue, we first need to construct suitable interaction records for students. Specifically, for student $s_i$, we randomly select one of his answered exercise $e_j \in E_i$ and flip the corresponding score on the interaction record (changing 1 to 0 or 0 to 1), and keep other records unchanged. By doing so, we generate the records for $s_{i'}$. Assume the proficiency of $s_{i'}$ is $\theta_{i,j}$, the fitting function for student $s_{i'}$ can be expressed as:

$$l_{i,j}(\theta_{i,j}) = - \sum_{k \in Q_i, k \neq j} [y_{i,k} \log p_k(\theta_{i,j}) + (1 - y_{i,k}) \log(1 - p_k(\theta_{i,j}))] \tag{3}$$
$$- [(1 - y_{i,j}) \log p_j(\theta_{i,j}) + y_{i,j} \log(1 - p_j(\theta_{i,j}))].$$

Next, we consider the partial order relationship between $s_i$, $s_{i'}$. Two scenarios need to be considered.

According to **Hypothesis 1**, when $\theta_i \geq \theta_{i,j}$ should hold but $\theta_i < \theta_{i,j}$ in fact, we need to add a regularization term $(\theta_{i,j} - \theta_i)^2$ to impose the constraint. Similarly, as for **Hypothesis 2**, when $\theta_i \leq \theta_{i,j}$ should hold but $\theta_i > \theta_{i,j}$ in fact, we need to add a regularization term $(\theta_i - \theta_{i,j})^2$ as penalty. As a result, we establish two regularization terms:

$$\Omega_1(\theta_i, \theta_{i,j}) = \mathbf{1}_{[\theta_i < \theta_{i,j}]}(\theta_{i,j} - \theta_i)^2, \quad \Omega_2(\theta_i, \theta_{i,j}) = \mathbf{1}_{[\theta_i > \theta_{i,j}]}(\theta_i - \theta_{i,j})^2, \tag{4}$$

where $\mathbf{1}$ is the indicator function. Then we summarize the final regularization term as:

$$\Omega(y_{i,j}, \theta_i, \theta_{i,j}) = y_{i,j}\Omega_1 + (1 - y_{i,j})\Omega_2. \tag{5}$$

Moreover, to enhance the monotonicity assumption, for each student $s_i$, we consider generating multiple fake students and comparing them with $s_i$ by randomly sample a subset $C_i \subset Q_i$. For each exercise in $C_i$, we perform a flipping operation on the corresponding score while keeping the other records unchanged, resulting in $|C_i|$ versions of student $s_i$.

As a result, the final loss function in CMCD is:

$$\mathcal{L}_{\mathcal{A}} = \frac{1}{m} \sum_{i=1}^{m} \left[ l_i(\theta_i) + \sum_{j \in C_i} (l_{i,j}(\theta_{i,j}) + c \cdot \Omega(y_{i,j}, \theta_i, \theta_{i,j})) \right], \tag{6}$$

where $c$ represents the strength of constraint.

## 5.2 Theoretical Guarantees

In this section, we provide theoretical guarantees of CMCD's effectiveness in terms of accuracy and convergence speed. For theoretical analysis, we will focus on the most classic cognitive diagnostic model, IRT (introduced in Section 3.2), which serves as the foundation for many other cognitive diagnostic models and has been widely implemented in GRE [30]. Please note that the following theoretical analysis can be easily extended to other CDMs.

Firstly, we analyze the accuracy of CMCD. Given that the core objective of cognitive diagnostic tasks is to estimate the true proficiency $\theta_i$ of student $s_i$, to assess the accuracy of our CMCD, we theoretically analyze the Maximum Likelihood Estimation (MLE) of $\theta_i$ through the optimization of $\mathcal{L}_{\mathcal{A}}$ (Eq. (6)). Let $\Theta = (\theta_1, \theta_2, \cdots, \theta_m)$, we assert that CMCD has accurate estimation stated as:

**Proposition 5.1** (Accuracy). *CMCD can accurately estimate the student proficiency level $\Theta$.*

To validate the above proposition, we prove that the MLE under CMCD is a consistent estimation:

**Theorem 5.2.** *Suppose a student $s_i$ with ability $\theta_i$ has records $R_i$ on exercises indexed by $Q_i$, and $\hat{\theta}_i^*$ denote the MLE of $\theta_i$ under $\mathcal{L}_{\mathcal{A}}$, then $\hat{\theta}_i^*$ is a consistent estimation of $\theta_i$, formally,*

$$\forall \varepsilon > 0, \delta > 0, \exists n_0 \in \mathbb{Z}^+, |Q_i| \geq n_0, \text{s.t.} \mathbf{P}\left(|\hat{\theta}_i^* - \theta_i| \geq \varepsilon\right) < \delta. \tag{7}$$

Detailed proofs for Theorem 5.2 can be found in Appendix A.1. The consistency estimation ensures the accuracy of CMCD. Next, we explore the superiority of CMCD in convergence speed compared to traditional CD models (the loss function $\mathcal{L}$ is introduced in Eq.(1)).

**Proposition 5.3** (Convergence Speed). *In terms of the estimation of proficiency level* $\Theta$*, CMCD has a faster convergence speed than the original approach.*

To validate the above proposition, we first notice that $\mathcal{L}$ and $\mathcal{L}_{\mathcal{A}}$ are both convex. Then, in the specific zone that violates the monotonicity assumption, we prove that $\mathcal{L}_{\mathcal{A}}$ are strong-convex in terms of $\Theta$. Finally we can analyze and compare the convergence speed in optimizations of $\mathcal{L}$ and $\mathcal{L}_{\mathcal{A}}$.

**Theorem 5.4.** $\mathcal{L}$ *and* $\mathcal{L}_{\mathcal{A}}$ *are convex in terms of* $\Theta$*. In addition,* $\mathcal{L}_{\mathcal{A}}$ *is strong-convex when*

$$\Theta \in D = \{(\theta_1, \theta_2, \ldots, \theta_m) | \forall i \in [m], \exists j \in C_i, y_{i,j} \mathbf{1}_{[\theta_i < \theta_{i,j}]} + (1 - y_{i,j}) \mathbf{1}_{[\theta_i > \theta_{i,j}]} = 1\}, \quad (8)$$

*and, meanwhile, we can also conclude* $\nabla^2 \mathcal{L}_{\mathcal{A}} > \nabla^2 \mathcal{L}$ *when* $\Theta \in D$*.*

Detailed proofs for Theorem 5.4 are provided in Appendices A.2. When performing the gradient descent algorithm to optimize the loss function, it is widely known that strong-convex functions converge faster than convex functions [3]. Overall, this represents the primary distinction between $\mathcal{L}_{\mathcal{A}}$ and $\mathcal{L}$. By examining the frequency of occurrences of this expression during the optimization process, we can analyze the convergence speeds of the two functions. The more frequent the occurrences, the faster the convergence of $\mathcal{L}_{\mathcal{A}}$. Simultaneously, this expression also indicates a violation of our hypotheses, necessitating the incorporation of regularization.

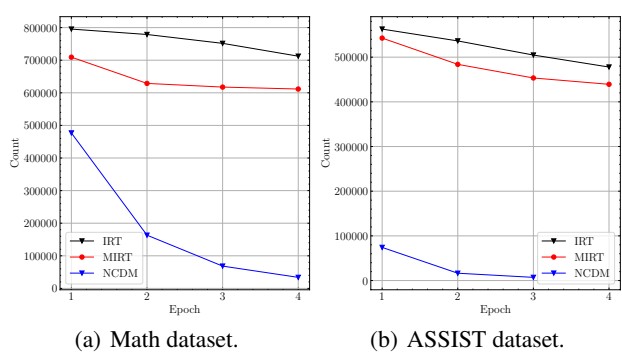

(a) Math dataset.   (b) ASSIST dataset.

Figure 5: The number of instances violating our hypotheses during the optimization process.

To validate the superior convergence speed, we calculated instances of hypothesis violation across different datasets, with the results illustrated in Figure 5. It reveals a prevalent occurrence of strong convexity in $\mathcal{L}_{\mathcal{A}}$ across various datasets and backbones. Therefore, by optimizing $\mathcal{L}_A$, CMCD converges faster.

## 6 Experiments

In this section, we first introduce the dataset and experimental setup. Then, we conduct extensive experiments on real-world datasets to answer the following questions:

- **RQ1:** Does CMCD learn fair and accurate cognitive diagnosis results?
- **RQ2:** How does the constraint of data augmentation impact CMCD?
- **RQ3:** Does CMCD have a faster convergence speed compared to other baseline models?
- **RQ4:** How do different hyper-parameter settings (i.e. $c$, $|C_i|$) affect the performance?

The code is released at https://github.com/Mercidaiha/CMCD.

### 6.1 Experimental Setup and Baselines

**Datasets** In this paper, we conduct experiments in two diagnostic datasets, namely Math and AS-SIST. The Math dataset was collected from an online educational system that provides homework, and evaluation for students. It collected records of junior high school students on mathematical exercises. In this dataset, we take the region as the sensitive attribute. The ASSIST dataset (ASSISTments 2009-2010 "skill builder") is an open dataset collected by the ASSISTments online tutoring systems [12], which includes student response logs and knowledge concepts [3]. Given that the ASSIST dataset does not provide sensitive attribute information, we follow [23] and categorize users into two groups based on whether the number of their responses exceeds 50. Regarding the dataset division, we allocate 80% of each student's response log for training and the remaining 20% for testing. The overview of basic statistics can be found in Appendix Table 2.

---

[3]https://sites.google.com/site/assistmentsdata

Table 1: The utility (U) and fairness (F) results on Math datasets. The best results on each backbone are highlighted in bold. ↓ indicates the lower, the better. ↑ indicates the higher, the better. We compare our CMCD with the Origin and get the p-value in the t-test. ($***: p < 0.001, **: p < 0.01, *: p < 0.05$.)

| | | IRT | | | | MIRT | | | | NCDM | | | |
|---|---|---|---|---|---|---|---|---|---|---|---|---|---|
| | Approach | RMSE↓ | MAE↓ | AUC↑ | ACC↑ | RMSE↓ | MAE↓ | AUC↑ | ACC↑ | RMSE↓ | MAE↓ | AUC↑ | ACC↑ |
| | Origin | 0.409 | 0.325 | 0.821 | 0.751 | 0.409 | 0.309 | 0.822 | 0.757 | 0.414 | 0.317 | 0.812 | 0.748 |
| | CD+Reg | 0.412 | 0.332 | 0.814 | 0.747 | 0.409 | 0.310 | 0.821 | 0.755 | 0.416 | 0.348 | 0.806 | 0.744 |
| | CD+EO | 0.412 | 0.334 | 0.813 | 0.744 | 0.413 | 0.317 | 0.814 | 0.750 | 0.420 | 0.321 | 0.803 | 0.740 |
| | CD+DP | 0.409 | 0.329 | 0.818 | 0.749 | 0.410 | 0.314 | 0.819 | 0.753 | 0.414 | 0.331 | 0.811 | 0.748 |
| U | CF-IRT | 0.409 | 0.314 | 0.821 | 0.753 | 0.411 | 0.304 | 0.823 | 0.755 | 0.420 | 0.316 | 0.804 | 0.743 |
| | CF-MIRT | 0.408 | 0.324 | 0.820 | 0.752 | 0.417 | 0.300 | 0.816 | 0.751 | 0.418 | 0.318 | 0.808 | 0.743 |
| | CF-NCDM | 0.406 | 0.318 | 0.824 | 0.756 | 0.406 | 0.312 | 0.826 | 0.758 | 0.417 | **0.312** | 0.809 | 0.747 |
| | CMCD | **0.394***** | **0.300***** | **0.842***** | **0.772***** | **0.406***** | **0.279***** | **0.834***** | **0.767***** | **0.413**** | 0.316** | **0.814***** | **0.749*** |
| | Approach | ΔRMSE↓ | ΔMAE↓ | ΔAUC↓ | ΔACC↓ | ΔRMSE↓ | ΔMAE↓ | ΔAUC↓ | ΔACC↓ | ΔRMSE↓ | ΔMAE↓ | ΔAUC↓ | ΔACC↓ |
| | Origin | 0.038 | 0.052 | 0.027 | 0.058 | 0.032 | 0.035 | 0.018 | 0.045 | 0.034 | 0.070 | 0.032 | 0.054 |
| | CD+Reg | 0.037 | 0.049 | 0.024 | 0.056 | 0.032 | 0.035 | 0.018 | 0.045 | 0.036 | 0.066 | 0.030 | 0.052 |
| | CD+EO | 0.041 | 0.041 | 0.022 | 0.062 | 0.036 | 0.025 | 0.020 | 0.047 | 0.040 | **0.054** | **0.027** | 0.057 |
| F | CD+DP | 0.038 | 0.059 | 0.033 | 0.059 | 0.033 | 0.036 | 0.020 | 0.043 | 0.035 | 0.076 | 0.032 | 0.050 |
| | CF-IRT | 0.038 | 0.042 | 0.023 | 0.055 | 0.034 | 0.040 | 0.013 | 0.047 | 0.042 | 0.056 | 0.039 | 0.056 |
| | CF-MIRT | 0.034 | 0.048 | 0.020 | 0.050 | 0.040 | 0.032 | 0.022 | 0.047 | 0.038 | 0.067 | 0.037 | 0.060 |
| | CF-NCDM | 0.034 | 0.044 | 0.022 | 0.050 | 0.031 | 0.042 | 0.018 | 0.043 | 0.039 | 0.055 | 0.034 | 0.052 |
| | CMCD | **0.025***** | **0.029***** | **0.009***** | **0.034***** | **0.028***** | **0.020***** | **0.005***** | **0.029***** | **0.033***** | 0.059*** | 0.028*** | **0.050** |

**Evaluation.** In this paper, we evaluate the results of user modeling by predicting scores. This evaluation can be divided into two aspects: utility and fairness evaluation. For utility evaluation, following previous works [39, 14], we adopt different metrics from the perspectives of both regression (MAE, RMSE) and classification (AUC, ACC). For fairness evaluation, we adopt the Definition introduced in Section 3.2. For the $\mathcal{M}$ in Eq. (2), we adopt MAE, RMSE, AUC, ACC (denoted as ΔMAE, ΔRMSE, ΔAUC, ΔACC in the experiment).

**Baseline approaches.** CMCD is a versatile framework applicable to various CD models. To validate its generality, we employed CMCD in three classical CD backbones, IRT [29], MIRT [5], NCDM [39]. For the baselines, we compare our methods with two categories of methods. Firstly, we compare our methods with the data augmentation baseline. To the best of our knowledge, there are currently no data augmentation methods specifically applied to diagnostics. Therefore, following the approach [44], we constructed the following baselines: CF-IRT, CF-MIRT, and CF-NCDM. Secondly, as our model enhances fairness, we compare it with several classical fairness baselines, such as CD+GF [23], CD+EO [47], CD+DP [47]. Further details regarding these baselines and the implementation of our methods can be found in Appendix A.3.

## 6.2 Experimental Results

**RQ1.** In this section, we discuss whether CMCD framework can alleviate the issue of data sparsity, thereby enhancing both the accuracy and fairness of the model. Specifically, we conducted comparative experiments with baselines on the Assistment and Math datasets, and the results of Math datasets results are presented in Table 1, the results of Assistment datasets can be found in Appendix A.4. From these results, we can draw the following findings:

(1) From the perspective of utility, we observed a significant enhancement in CMCD across different datasets and various backbones. This underscores the effectiveness of our approach. Furthermore, we noted that, compared to backbone models like NCDM that rely on neural networks, CMCD demonstrates more pronounced improvements on traditional backbones (IRT). We attribute this to the fact that models dependent on neural network backbones, such as NCDM, exhibit superior data fitting capabilities, thereby alleviating issues related to data sparsity. Consequently, the gains achieved by CMCD on NCDM are less conspicuous. In contrast, traditional IRT models are more susceptible to the impact of data sparsity, making CMCD's influence more substantial in such cases.

(2) From the perspective of fairness, we observed that CMCD can, to a certain extent, alleviate unfair phenomena in cognitive diagnostic models. Particularly noteworthy is its performance on the Math dataset, where our model achieved the best fairness outcomes across different backbones (IRT, MIRT). This underscores that CMCD, while mitigating data sparsity issues, concurrently promotes fairness in model outcomes.

(3) From the perspective of the trade-off between fairness and utility, our model consistently surpasses the baselines with data augmentation in both performance and fairness outcomes. While compared to fairness-aware baselines, our model may not always outperform dedicated fairness-enhancing methods on specific datasets, it is important to note that these baselines often come at the expense of utility. In contrast, CMCD not only enhances fairness but also improves performance. This highlights that CMCD achieves the optimal trade-off between fairness and utility.

**RQ2.** To address the issue of data sparsity, we introduce two hypotheses for data augmentation. In this section, we validate the effectiveness of these two hypotheses. Specifically, we conduct ablation experiments on the Math dataset, where we systematically removed the corresponding hypothesis strategies. The results are illustrated in Figure 6. It is evident that after removing different hypothesis strategies, both Fairness and Utility performances exhibited varying degrees of decline, providing evidence for the efficacy of each strategy. Additionally,

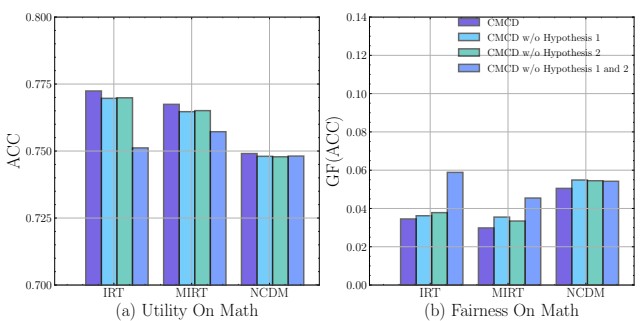

Figure 6: The ablation experiment on Math dataset.

we observed that these two hypotheses yield different effects on various backbones. In comparison to NCDM, a neural network-dependent model, hypotheses perform better on traditional models like IRT and MIRT. We attribute this to the powerful data-fitting capability of NCDM, which alleviates issues related to data sparsity. In contrast, models such as IRT and MIRT, constrained by data sparsity, exhibit limited capabilities.

**RQ3.** In Section 5.2, we theoretically demonstrated that CMCD exhibits a faster convergence rate. In this section, we empirically validate the superiority of CMCD in terms of convergence speed. Specifically, we compared the convergence speeds (the epoch at which early stopping occurs) of different baselines on various backbones using the Math dataset, as depicted in Figure 7. It is observed that in the early stages, CMCD shows minimal differences compared to other baselines. We attribute this to the fact

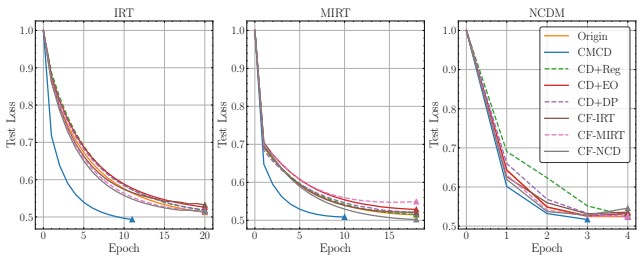

Figure 7: Convergence against training epoch for different backbones on Math dataset.

that, initially, the effectiveness of the backbone models is not optimal, resulting in less precise enhancement of the monotonicity assumption. However, as the number of epochs increases, CMCD significantly outpaces other baselines, demonstrating a faster convergence.

**RQ4.** In CMCD, two hyperparameters control the effectiveness. Specifically, $c$ controls the strength of the constraints, and $|C_i|$ regulates the number of generated students. In this section, we investigate the impact of adjusting these two hyperparameters on CMCD using the Math dataset. The specific results are presented in Figure 8 and Figure 9. In Figure 8, a consistent trend is observed across different backbones. Initially, with the strengthening of $c$, both fairness and utility show improvement, indicating the efficacy of our monotonicity enhancement. However, as $c$ reaches a certain intensity, both fairness and utility experience a decline. We attribute this to the possibility that an excessively large $c$ might interfere with the model's normal training, leading to a decrease in effectiveness. Moving to Figure 9, diverse trends are identified across different backbones. Specifically, in the case of IRT, as the number of contrasts increases, both fairness and utility consistently improve. Conversely, for NCDM, as the quantity of contrasts grows, there is a continuous decline in utility. We attribute these observations to inherent differences in the nature of the models. IRT is more affected by data sparsity, so enhancing $C_i$ effectively mitigates data sparsity, yielding improved results. On

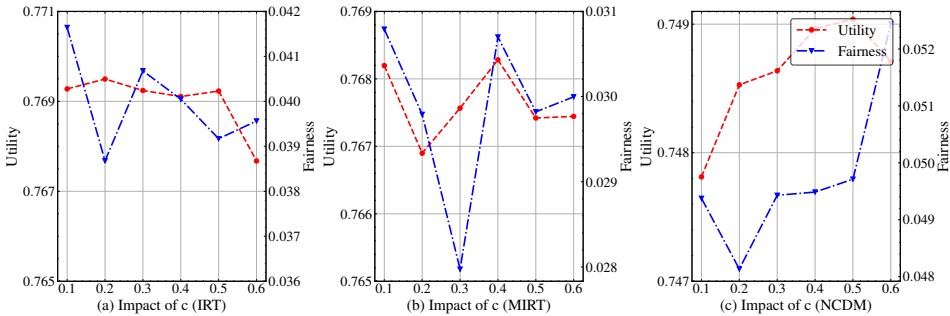

Figure 8: The impact of $c$ for different CD backbones on the Math. For utility, the higher the better; For fairness, the lower the better.

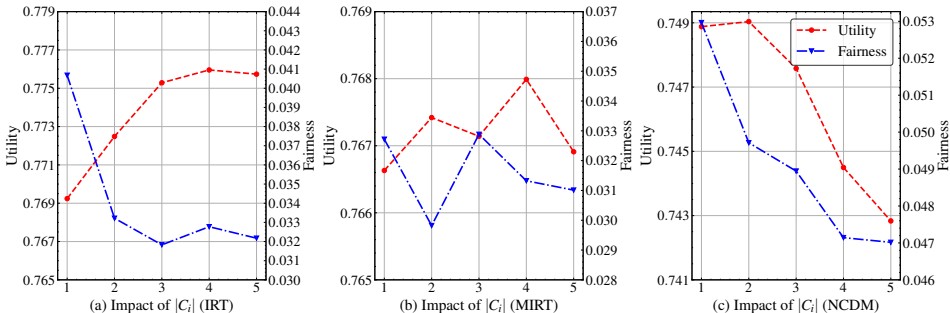

Figure 9: The impact of $|C_i|$ for different CD backbones on the Math. For utility, the higher the better; For fairness, the lower the better.

the other hand, NCDM, benefiting from its neural network's fitting capacity, partially alleviates data sparsity issues. Consequently, an increase in contrasts may interfere with its intrinsic performance.

## 7 Conclusion and Discussion

In this paper, we conducted a focused study on addressing data sparsity in CD. Initially, we empirically validated that data sparsity leads to inaccurate and unfair diagnostic results. Subsequently, by integrating data augmentation and the monotonicity assumption, we introduced two constraints to alleviate the issue of data sparsity. Moreover, we provided theoretical guarantees regarding accuracy and convergence speed. The experiments on real-world datasets demonstrated the effectiveness of our approach in addressing the data sparsity issue, culminating in fair and accurate diagnosis results.

CD is a highly significant task in intelligent education. The generated diagnostic results can be applied to various areas, e.g., personalized tutoring and intelligent question recommendations. This can reduce the burden on teachers and students, providing effective learning experiences for students. Our CMCD mitigates the unfairness and inaccuracy issues arising from data sparsity, thereby promoting educational equity to a certain extent. Simultaneously, we recognize that CD relies on students' response records that might be inaccessible due to privacy concerns. In the future, we will contemplate incorporating federated learning to enhance diagnostic services and ensures student privacy.

## Acknowledgments and Disclosure of Funding

This research was supported by grants from the National Key Research and Development Program of China (Grant No. 2021YFF0901003), the Key Technologies R&D Program of Anhui Province (No. 202423k09020039) and the Fundamental Research Funds for the Central Universities.

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

# A Appendix

## A.1 Proof of Theorem 5.2.

**Theorem A.1.** *Suppose a student $s_i$ with ability $\theta_i$ has records $R_i$ on exercises indexed by $Q_i$, and $\hat{\theta}_i^*$ denote the MLE of $\theta_i$ under $\mathcal{L}_A$, then $\hat{\theta}_i^*$ is a consistent estimation of $\theta_i$, formally,*

$$\forall \varepsilon > 0, \delta > 0, \exists n_0 \in \mathbb{Z}^+, |Q_i| \geq n_0, \text{s.t.} \mathbf{P}\left(|\hat{\theta}_i^* - \theta_i| \geq \varepsilon\right) < \delta. \tag{9}$$

*Proof.* We denote $\hat{\theta}_i$ as the maximum likelihood estimate (MLE) of $\theta_i$ for the loss function $\mathcal{L}$ (Eq.(1)). The mean negative log-likelihood function of student $s_i$ is

$$l_i(\theta) = -\sum_{j \in Q_i} [y_{i,j} \log p_j(\theta) + (1 - y_{i,j}) \log(1 - p_j(\theta))] \tag{10}$$

We can further validate that

$$
\begin{aligned}
l_i'(\theta) &= -\sum_{j \in Q_i} [\alpha_j (y_{i,j} - p_j(\theta))], \\
l_i''(\theta) &= \sum_{j \in Q_i} \alpha_j^2 p_j(\theta)(1 - p_j(\theta)).
\end{aligned}
\tag{11}
$$

On the other hand, the MLE $\hat{\theta}_i$ satisfies $l_i'(\hat{\theta}_i) = 0$, by taking Taylor expansion,

$$0 = l_i'(\hat{\theta}_i) = l_i'(\theta_i) + l_i''(\xi_i)(\hat{\theta}_i - \theta_i), \tag{12}$$

where $\xi_i$ is between $\hat{\theta}_i$ and $\theta_i$. We can further get

$$\left|\hat{\theta}_i - \theta_i\right| = \left|-\frac{l_i'(\theta_i)}{l_i''(\xi_i)}\right| = \left|\frac{\sum_{j \in Q_i} \alpha_j (y_{i,j} - p_j(\theta_i))}{\sum_{j \in Q_i} \alpha_j^2 p_j(\xi_i)(1 - p_j(\xi_i))}\right| \tag{13}$$

We can see that $\{y_{i,j} : j \in Q_i\}$ are independent and $\mathbf{E}[y_{i,j}] = p_j(\theta_i)$. Through Chebyshev's Law of Large Numbers, $\forall \varepsilon > 0, \delta > 0$, there exists $n_0$ such that

$$\mathbf{P}\left[\left|\frac{1}{|Q_i|}\sum_{j \in Q_i} \alpha_j y_{i,j} - \frac{1}{|Q_i|}\sum_{j \in Q_i} \alpha_j p_j(\theta_i)\right| \geq \frac{1}{4}\alpha^2 \varepsilon\right] < \delta, \tag{14}$$

when $|Q_i| \geq n_0$ holds, here $\alpha = \max_j \alpha_j$. Considering

$$\frac{1}{|Q_i|} \sum_{j \in Q_i} \alpha_j^2 p_j(\xi_i)(1 - p_j(\xi_i)) \leq \frac{1}{4|Q_i|} \sum_{j \in Q_i} \alpha_j^2 \leq \frac{1}{4|Q_i|} \sum_{j \in Q_i} \alpha^2 = \frac{1}{4}\alpha^2, \quad (15)$$

we can see when $|Q_i| \geq n_0$, with probability less than $\delta$,

$$\left| \hat{\theta}_i - \theta_i \right| \geq \frac{\alpha^2 \varepsilon / 4}{\alpha^2 / 4} = \varepsilon, \quad (16)$$

which means $\hat{\theta}_i$ is a consistent estimation of $\theta_i$. Then, we denote $\hat{\theta}_{i,j} = \arg\min_\theta l_{i,j}(\theta)$, and we have

$$l'_{i,j}(\hat{\theta}_i) = l'_i(\hat{\theta}_i) + (l_{i,j} - l_i)'(\hat{\theta}_i) = -\alpha_j(1 - 2y_{i,j}). \quad (17)$$

If $y_{i,j} = 1$, $l'_{i,j}(\hat{\theta}_i) = \alpha_j > 0$ and it means $\hat{\theta}_i$ is bigger than the zero point, i.e., $\hat{\theta}_i > \hat{\theta}_{i,j}$. If $y_{i,j} = 0$, $l'_{i,j}(\hat{\theta}_i) = -\alpha_j < 0$ and it means $\hat{\theta}_i$ is smaller than the zero point, i.e., $\hat{\theta}_i < \hat{\theta}_{i,j}$. Overall, we can check that $\Omega(y_{i,j}, \hat{\theta}_i, \hat{\theta}_{i,j}) = 0$.

In this way, $\hat{\theta}_i$ makes $l_i(\theta)$ the smallest, $\hat{\theta}_{i,j}$ makes $l_{i,j}(\theta)$ the smallest, and with them the regularized terms are always 0. Therefore, such $\hat{\theta}_i$ and $\hat{\theta}_{i,j}$ are the minimum points of $\mathcal{L}_A$, and this fact results in $\hat{\theta}_i^* = \hat{\theta}_i$. Combining the conclusion that $\hat{\theta}_i$ is a consistent estimation of $\theta_i$, we can conclude that $\hat{\theta}_i^*$ is a consistent estimation of $\theta_i$. $\qquad\square$

## A.2 Proof of Theorem 5.4.

**Theorem A.2.** *$\mathcal{L}$ and $\mathcal{L}_A$ are convex in terms of $\Theta$. In addition, $\mathcal{L}_A$ is strong-convex when*

$$\Theta \in D = \{(\theta_1, \theta_2, \ldots, \theta_m) | \forall i \in [m], \exists j \in C_i, y_{i,j}\mathbf{1}_{[\theta_i < \theta_{i,j}]} + (1 - y_{i,j})\mathbf{1}_{[\theta_i > \theta_{i,j}]} = 1\}, \quad (18)$$

*and, meanwhile, we can also conclude $\nabla^2 \mathcal{L}_A > \nabla^2 \mathcal{L}$ when $\Theta \in D$.*

*Proof.* For the loss function $\mathcal{L}$ (Eq.(1)), the mean negative log-likelihood function of student $s_i$ is

$$l_i(\theta) = -\sum_{j \in Q_i} [y_{i,j} \log p_j(\theta) + (1 - y_{i,j}) \log(1 - p_j(\theta))] \quad (19)$$

We can further validate that

$$l'_i(\theta) = -\sum_{j \in Q_i} [\alpha_j(y_{i,j} - p_j(\theta))], \quad l''_i(\theta) = \sum_{j \in Q_i} \alpha_j^2 p_j(\theta)(1 - p_j(\theta)) \geq 0. \quad (20)$$

For any $i, j$ we have $\frac{\partial^2 \mathcal{L}}{\partial \theta_i^2} \geq 0$ and $\frac{\partial^2 \mathcal{L}}{\partial \theta_i \partial \theta_j} = 0$, hence $\nabla^2 \mathcal{L} \geq 0$ and $\mathcal{L}$ is convex in terms of $\Theta$.

In addition, we can see that

$$\frac{\partial^2 \mathcal{L}_A}{\partial \theta_i^2} = \frac{\partial^2 \mathcal{L}}{\partial \theta_i^2} + 2c \sum_{j \in C_i} \left(y_{i,j}\mathbf{1}_{[\theta_i < \theta_{i,j}]} + (1 - y_{i,j})\mathbf{1}_{[\theta_i > \theta_{i,j}]}\right) \geq \frac{\partial^2 \mathcal{L}}{\partial \theta_i^2} \geq 0, \quad (21)$$

and $\frac{\partial^2 \mathcal{L}}{\partial \theta_i \partial \theta_j} = 0$ for all $i, j$. Hence $\nabla^2 \mathcal{L}_A \geq \nabla^2 \mathcal{L} \geq 0$ and $\mathcal{L}_A$ is also convex in terms of $\Theta$.

When $\Theta \in D$, we can see that

$$2c \sum_{j \in C_i} \left(y_{i,j}\mathbf{1}_{[\theta_i < \theta_{i,j}]} + (1 - y_{i,j})\mathbf{1}_{[\theta_i > \theta_{i,j}]}\right) > 0. \quad (22)$$

Based on Eq.(21), we can conclude that $\mathcal{L}_A$ is strong-convex and $\nabla^2 \mathcal{L}_A > \nabla^2 \mathcal{L}$ when $\Theta \in D$. In fact, we have $\nabla^2 \mathcal{L} \geq 2c \cdot \min_i\{\sum_{j \in C_i} y_{i,j}\mathbf{1}_{[\theta_i < \theta_{i,j}]} + (1 - y_{i,j})\mathbf{1}_{[\theta_i > \theta_{i,j}]}\}$ when $\Theta \in D$. $\qquad\square$

Table 2: The statistics of the datasets.

|  | Math | ASSIST |
|---|---|---|
| #Students | 7,177 | 4,163 |
| #Exercises | 12,129 | 17,746 |
| #Knowledge concepts | 2,076 | 123 |
| #Response logs | 384,356 | 324,572 |
| Sparsity | 99.558% | 99.561% |

## A.3 Experiment Details

**Baseline approaches.** CMCD is a versatile framework applicable to various CD models. To validate its generality, we employed CMCD in three classical CD backbones, IRT [29], MIRT [5], NCDM [39]. The details of baselines are as follows:

- IRT [29] is a cognitive diagnosis method that models the cognitive processes from students' exercise records with a logistic-like function.
- MIRT [5] is a variant of the basic IRT model, which extends the unidimensional latent traits of students and exercises in IRT to multidimensional vectors.
- NCDM [39] is a deep neural cognitive diagnosis framework that models the complex interaction from students' exercising records with a multilayer perceptron (MLP).

CMCD can alleviate the issue of data sparsity in diagnostic models, enhancing both accuracy and fairness. To validate the effectiveness of our approach, we compare it with two categories of methods. The first category is data augmentation methods. To the best of our knowledge, there are currently no data augmentation methods specifically applied to diagnostics. Therefore, we adapt the data augmentation concept "What would ... if ...?" from traditional recommendation systems to the cognitive diagnostic field. Specifically, following the approach in [44], we apply a causal rule: "What would the student's response be if they encountered a new question?" Additionally, considering the characteristics of CD domain, we make appropriate adaptations by using classic IRT, MIRT, and NCDM models to predict responses to new questions. Ultimately, we extend the data augmentation baseline to CF-IRT, CF-MIRT, and CF-NCDM. The second category is fairness-aware methods. Since our model enhances fairness, we compare it with several classical fairness baselines, CD+GF [23], CD+EO [47], CD+DP [47]. The details of baselines are as follows:

- CD+GF: a well-known fairness improvement strategy that considers the fairness metric as a regularization for the loss and has been used in prior fairness works [47, 23]. In our work, we use Eq. (4) as a regularization for Eq. (2);
- CD+EO: a method regards Equal opportunity (EO) as a regularization [16].
- CD+DP: a method regards Demographic Parity (DP) as a regularization [16].

**Implementation detail.** In terms of model parameter configuration, NCDM adheres to the settings outlined in [39]. The fully connected layers have dimensions of 512, 256, and 1. The sigmoid function serves as the activation function for all layers. MIRT's dimension of student proficiency parameters matches those in NCDM. For all models, we set the learning rate to 0.001 and the dropout rate to 0.2. We apply Adam as the optimization algorithm to update the model parameters. We implement all models with PyTorch and conduct all experiments on four 2.0GHz Intel Xeon E5-2620 CPUs and a Tesla K20m GPU.

## A.4 The results on Assistment dataset (RQ1)

The results for the Assistment datasets can be found in Table 3. They exhibit similar patterns as those introduced in RQ1.

Table 3: The utility (U) and fairness (F) results on Assistment datasets. The best results on each backbone are highlighted in bold. ↓ indicates the lower, the better. ↑ indicates the higher, the better. We compare our CMCD with the Origin and get the p-value in the t-test. ($***: p < 0.001, **: p < 0.01, *: p < 0.05$.)

| | Approach | IRT | | | | MIRT | | | | NCDM | | | |
|---|---|---|---|---|---|---|---|---|---|---|---|---|---|
| | | RMSE↓ | MAE↓ | AUC↑ | ACC↑ | RMSE↓ | MAE↓ | AUC↑ | ACC↑ | RMSE↓ | MAE↓ | AUC↑ | ACC↑ |
| **U** | Origin | 0.449 | 0.372 | 0.707 | 0.693 | 0.447 | 0.344 | 0.733 | 0.707 | 0.433 | 0.352 | 0.745 | 0.726 |
| | CD+Reg | 0.446 | 0.375 | 0.713 | 0.695 | 0.451 | 0.355 | 0.719 | 0.698 | **0.433** | 0.372 | 0.742 | 0.725 |
| | CD+EO | 0.447 | 0.378 | 0.708 | 0.693 | 0.449 | 0.348 | 0.725 | 0.706 | 0.433 | 0.366 | 0.742 | 0.725 |
| | CD+DP | 0.451 | 0.389 | 0.693 | 0.685 | 0.450 | 0.350 | 0.723 | 0.701 | 0.434 | 0.362 | 0.741 | 0.721 |
| | CF-IRT | 0.448 | 0.381 | 0.704 | 0.691 | 0.449 | 0.340 | 0.730 | 0.709 | 0.434 | 0.339 | 0.748 | 0.726 |
| | CF-MIRT | 0.447 | 0.372 | 0.712 | 0.695 | 0.449 | 0.338 | 0.733 | 0.707 | 0.434 | 0.343 | 0.747 | 0.724 |
| | CF-NCDM | 0.448 | 0.377 | 0.707 | 0.693 | 0.448 | 0.344 | 0.732 | 0.705 | 0.436 | 0.336 | 0.748 | 0.725 |
| | CMCD | **0.437\*\*\*** | **0.346\*\*\*** | **0.745\*\*\*** | **0.715\*\*\*** | **0.432\*\*\*** | **0.337\*\*\*** | **0.758\*\*\*** | **0.724\*\*\*** | 0.439\*\*\* | **0.326\*** | **0.751\*\*** | **0.727\*** |
| | Approach | ΔRMSE↓ | ΔMAE↓ | ΔAUC↓ | ΔACC↓ | ΔRMSE↓ | ΔMAE↓ | ΔAUC↓ | ΔACC↓ | ΔRMSE↓ | ΔMAE↓ | ΔAUC↓ | ΔACC↓ |
| **F** | Origin | 0.035 | 0.043 | 0.028 | 0.057 | 0.039 | 0.042 | 0.032 | 0.052 | 0.019 | 0.026 | 0.006 | 0.027 |
| | CD+Reg | 0.035 | 0.041 | 0.025 | 0.056 | 0.036 | 0.036 | **0.021** | 0.050 | 0.018 | 0.025 | 0.007 | 0.029 |
| | CD+EO | 0.035 | **0.031** | **0.018** | 0.064 | 0.038 | **0.024** | 0.032 | **0.044** | 0.020 | 0.023 | 0.005 | 0.030 |
| | CD+DP | 0.034 | 0.048 | 0.031 | 0.060 | 0.036 | 0.038 | 0.022 | 0.049 | 0.021 | 0.038 | 0.003 | 0.039 |
| | CF-IRT | 0.038 | 0.041 | 0.027 | 0.060 | 0.045 | 0.048 | 0.040 | 0.060 | 0.019 | 0.027 | 0.004 | 0.025 |
| | CF-MIRT | 0.037 | 0.041 | 0.027 | 0.060 | 0.043 | 0.040 | 0.032 | 0.053 | **0.017** | 0.025 | 0.410 | 0.026 |
| | CF-NCDM | 0.035 | 0.042 | 0.027 | 0.057 | 0.044 | 0.058 | 0.048 | 0.062 | 0.018 | 0.026 | 0.005 | 0.024 |
| | CMCD | **0.031\*\*\*** | **0.042\*\*** | **0.027\*\*\*** | **0.042\*\*\*** | **0.035\*** | **0.036\*\*** | 0.027 | **0.048\*\*** | **0.018\*\*\*** | **0.023\*** | **0.002\*\*\*** | **0.021** |

