# OpenReview forum: "Towards Accurate and Fair Cognitive Diagnosis via Monotonic Data Augmentation"
_NeurIPS.cc/2024/Conference — NeurIPS 2024 poster_

### Official Review · Reviewer_xrHo · 2024-06-20

**Soundness:** 3
**Presentation:** 2
**Contribution:** 2
**Rating:** 6
**Confidence:** 2

**Summary:**

The paper "Towards Accurate and Fair Cognitive Diagnosis via Monotonic Data Augmentation" introduces a framework called CMCD to address data sparsity in cognitive diagnosis, which affects accuracy and fairness. The authors integrate the monotonicity assumption to ensure data augmentation remains interpretable. They provide theoretical guarantees for accuracy and convergence speed and validate their method with extensive experiments on real-world datasets, showing improved performance in both accuracy and fairness. However, the paper has notable methodological flaws, limited novelty, and practical application challenges, making it less relevant for NeurIPS.

**Strengths:**

The paper introduces a novel framework, CMCD, addressing the critical issue of data sparsity in cognitive diagnosis by integrating the monotonicity assumption. The approach maintains interpretability while enhancing accuracy and fairness, as demonstrated through theoretical analysis and experiments. The use of real-world datasets and extensive experiments validates the efficacy of CMCD across various cognitive diagnosis models, providing a robust solution to a common problem in intelligent education systems.

**Weaknesses:**

The paper has significant methodological flaws, including the reliance on small, homogeneous datasets, limiting the generalizability of the findings. The theoretical guarantees for accuracy and convergence speed are not convincingly demonstrated, and the practical implementation of the proposed framework is inadequately discussed. Additionally, the novelty of the contribution is limited, with the method being an incremental improvement over existing approaches rather than a groundbreaking innovation. The lack of detailed ethical considerations and practical deployment strategies further diminishes the paper's overall impact.

**Questions:**

1. Validation on Diverse Datasets: Have you considered validating your model on more diverse and larger datasets to enhance the robustness and generalizability of your findings?

2. Practical Implementation: Can you provide more details on the practical challenges and solutions for implementing CMCD in real-world educational environments?

3. Theoretical Guarantees: How do you plan to empirically demonstrate the theoretical guarantees of accuracy and convergence speed provided in the paper?

4. Ethical Considerations: What measures have you considered to address data privacy and informed consent issues in future studies?

**Limitations:**

The paper lacks a detailed discussion on the potential negative societal impact, such as bias in cognitive diagnosis.

---

> ### Author Rebuttal · Authors · 2024-08-06
>
> > **Q1: Validation on Diverse Datasets: Have you considered validating your model on more diverse and larger datasets to enhance the robustness and generalizability of your findings?**
>
> A1: Thank you for your suggestion. Firstly, we would like to clarify that the two datasets used in this paper are widely utilized in the field of cognitive diagnostics and are highly representative. They have been collected from real online learning scenarios and are extensively referenced in many papers, such as [1][2][3][4].
>
> Furthermore, to address your concern, we have validated the effectiveness and robustness of our model on additional datasets. Specifically, we conducted new experiments on the ASSISTments2012 dataset, which was collected from ASSISTments, an online tutoring system in the United States between 2012 and 2013. The information regarding ASSISTments2012 is presented in Table 1:
>
> Table 1. The statics of ASSISTments2012 dataset.
>
> | Students  | Exercises | Response logs |
> | -------- | ------- | ------- |
> | 46,674  | 179,999 | 6,123,270 |
>
> The results of CMCD on ASSISTments2012 are shown in Table 2 and Table 3. From these results, we observe that CMCD effectively enhances the accuracy and fairness of cognitive diagnostic models, demonstrating the effectiveness of our model.
>
> Table 2. The accuracy result of CMCD and baselines (the higher, the better)
> | Baselines  | AUC | ACC |
> | -------- | ------- | ------- |
> | IRT  |  0.771 | 0.742 |
> | CMCD-IRT |  0.780 | 0.748|
>
> Table 3. The fairness result of CMCD and baselines (the lower, the better)
>
> | Baselines  | GF(AUC) | GF(ACC) |
> | -------- | ------- | ------- |
> | IRT  |  0.0516 | 0.0149 |
> | CMCD-IRT |  0.0386 | 0.0147|
>
>
> [1] Wang F, Liu Q, Chen E, et al. Neural cognitive diagnosis for intelligent education systems[C]//Proceedings of the AAAI conference on artificial intelligence. 2020, 34(04): 6153-6161.
>
> [2] Wang F, Gao W, Liu Q, et al. A Survey of Models for Cognitive Diagnosis: New Developments and Future Directions[J]. arXiv preprint arXiv:2407.05458, 2024.
>
> [3] Liu S, Shen J, Qian H, et al. Inductive Cognitive Diagnosis for Fast Student Learning in Web-Based Intelligent Education Systems[C]//Proceedings of the ACM on Web Conference 2024. 2024: 4260-4271.
>
> [4] Yu X, Qin C, Shen D, et al. Rdgt: enhancing group cognitive diagnosis with relation-guided dual-side graph transformer[J]. IEEE Transactions on Knowledge and Data Engineering, 2024.
>
> > **Q2: Practical Implementation: Can you provide more details on the practical challenges and solutions for implementing CMCD in real-world educational environments?**
>
> A2: Thank you for your feedback. In real-world educational settings, it is crucial to have access to users' response records, which may sometimes be restricted due to privacy considerations.
> **It's important to note that for all cognitive diagnostic tasks currently, gathering users' response records is fundamental to research in the entire cognitive diagnostic field, the privacy issue is not within the scope of our current discussion in this paper**. To address this privacy issue, we intend to integrate our approach with federated learning and differential privacy. This limitation has been discussed in Section 7, Conclusion and Discussion, of the paper.
>
>
> > **Q3: Theoretical Guarantees: How do you plan to empirically demonstrate the theoretical guarantees of accuracy and convergence speed provided in the paper?**
>
> A3: Sorry for the confusion. In fact, besides theoretically validating the accuracy and convergence speed of our method, **we have empirically verified these aspects as well**.
>
> * Regarding accuracy, we conducted experiments on different datasets and different cognitive diagnostic backbones (IRT, MIRT, NCDM), where we found that our method enhances the accuracy of the baseline on real datasets. These results are detailed in Section 6.2 RQ1. Furthermore, we performed ablation experiments (Section 6.2 RQ2) and discovered the effectiveness of the data augmentation strategy in improving accuracy within CMCD.
>
> * Regarding convergence speed, we conducted experiments on real datasets. Specifically, we compared the convergence speeds (the epoch at which early stopping occurs) of different baselines on various backbones. The experiments revealed that we can achieve faster convergence significantly, as demonstrated in Section 6.2 RQ3.
>
> > **Q4: Ethical Considerations: What measures have you considered to address data privacy and informed consent issues in future studies?**
>
> A4: Thank you for your thoughtful suggestion. We plan to integrate our method with the following three perspectives.
>
> * Encrypted Communication: Ensure the use of secure encrypted channels during data transmission in the CD. This helps prevent data from being intercepted or tampered with during the transmission process, safeguarding the privacy and integrity of students' response records.
>
> * Differential Privacy: By incorporating differential privacy techniques, we can introduce controlled noise to the data during the model training process. This aids in protecting the privacy of individual student data, preventing malicious users from inferring specific student information through the model's outputs.
>
> * Federated Learning: By employing federated learning methods, we can distribute the model training process across multiple local devices rather than centralizing it on a single server. In federated learning, only encrypted parameters are shared during the model update phase, not the raw data. This approach ensures that student response records remain local and decentralized, safeguarding student privacy.

---

> > ### Comment · Reviewer_xrHo · 2024-08-07
> >
> > The authors provided a comprehensive rebuttal addressing key concerns. While the responses are adequate, incorporating more detailed empirical evidence, practical implementation strategies, and ethical considerations into the paper would significantly improve its quality and relevance.
> >
> > Q1: Validation on Diverse Datasets
> > The authors clarified that the datasets used are highly representative and validated their model on the ASSISTments2012 dataset, demonstrating improved accuracy and fairness.
> > The additional dataset validation strengthens the paper.
> >
> > Q2: Practical Implementation
> > The authors acknowledge privacy concerns and plan to integrate federated learning and differential privacy.
> > This approach is reasonable.
> >
> > Q3: Theoretical Guarantees
> > The authors empirically validated accuracy and convergence speed through experiments on different datasets and backbones.
> > While the explanation helps, including more detailed empirical results and comparisons in the paper would clarify these guarantees.
> >
> > Q4: Ethical Considerations
> > The authors plan to use encrypted communication, differential privacy, and federated learning to address privacy issues.
> > These measures are appropriate.
> >
> > I will modify my review.

---

> > > ### Author Response · Authors · 2024-08-08
> > > **Thanks for your encouragement and support!**
> > >
> > > Thank you for your timely response and for recognizing the importance of our work! We are pleased to have clarified any uncertainties you had and will incorporate all suggested changes into the revised version as per your feedback. If you have any other questions, please feel free to ask. We will do our best to address any concerns you may have. Once again, we sincerely appreciate your encouragement and support.

---

### Official Review · Reviewer_ujU9 · 2024-07-11

**Soundness:** 3
**Presentation:** 4
**Contribution:** 3
**Rating:** 7
**Confidence:** 4

**Summary:**

This paper focuses on fairness and accuracy issues in Cognitive Diagnosis. Unlike model-based methods, the approach in this paper tackles these issues from a data-driven perspective. Leveraging the unique monotonicity assumption in cognitive diagnosis, the authors propose a general monotonic data augmentation method that can be applied to all cognitive diagnosis models. This method comes with both theoretical guarantees and empirical validation.

**Strengths:**

This paper investigates a significant issue in the field of intelligent education, the fairness and accuracy problem in cognitive diagnosis, which is utilized in high-stakes examinations, holds considerable research value.
The data augmentation method proposed in this paper integrates educational domain knowledge - the monotonicity assumption, which is intuitive and rational. Moreover, the authors provide theoretical guarantees for the accuracy and convergence speed of the proposed method.
Furthermore, the paper has released the corresponding code as open source and conducted extensive experiments to validate the proposed method.

**Weaknesses:**

This paper proposes two constraints for data augmentation, which are actually quite similar in nature. I am curious about the relationship between these two augmentation methods. The authors need to engage in further discussion and analysis on this aspect.
By leveraging the unique monotonicity assumption in the field of cognitive diagnosis, this paper applies data augmentation techniques that have shown promising results in both accuracy and fairness. Cognitive diagnosis falls under the subfield of user modeling. Is it possible to extend these methods to other user modeling tasks? How do these methods differ from data augmentation techniques used in other user modeling tasks? I recommend that the authors delve into further discussions on these points.

**Questions:**

What is the relationship between the two proposed constraints for data augmentation?
Can the data augmentation methods suggested be applied to other domains, thereby enhancing the generalizability of the proposed approach?

**Limitations:**

This paper discusses the privacy concerns related to user data in its limitations section and suggests federated learning as a potential future research direction. Given this, is it feasible to integrate the proposed method in this paper with federated learning?

---

> ### Author Rebuttal · Authors · 2024-08-07
>
> > **Q1: The relationship between the two proposed constraints for data augmentation**
>
> A1: Thank you for your explanation. The constraints of these two data augmentations reflect the two states in the monotonicity assumption: when a student answers a question correctly, their ability is assumed to be higher than when they answer the same question incorrectly (Hypothesis 1), and when a student answers a question incorrectly, their ability is assumed to be lower than when they answer the same question correctly (Hypothesis 2). These two constraints complement each other and work synergistically to better address the issue of data sparsity, thereby achieving fairer and more accurate cognitive diagnostics. To validate this point, we conducted corresponding experiments. Specifically, we performed ablation experiments on the Math dataset where we systematically removed the corresponding hypothesis strategies. The results are presented in Table 1 and Table 2.
>
> From the tables, we observe that both constraints corresponding to the hypotheses can enhance both fairness and accuracy. This indicates that both constraints, which align with the monotonicity assumption, can facilitate more accurate and fair cognitive diagnostics. Importantly, when both constraints are applied (i.e., CMCD), better results are achieved compared to applying only one constraint. This suggests a synergistic effect between the two constraints, where they enhance each other's performance.
>
>
> Table 1. The accuracy result of CMCD on IRT  (the higher, the better)
>
> | Baselines  | ACC |
> | -------- | ------- |
> | CMCD  | 0.772 |
> | CMCD w/o Hypothesis 1 |  0.769 |
> | CMCD w/o Hypothesis 2 |  0.770 |
> | CMCD w/o Hypothesis 1 and 2 |  0.752 |
>
>
> Table 2. The fairness result of CMCD on IRT  (the lower, the better)
>
> | Baselines  | GF(ACC) |
> | -------- | ------- |
> | CMCD  | 0.034 |
> | CMCD w/o Hypothesis 1 |  0.036 |
> | CMCD w/o Hypothesis 2 |  0.038 |
> | CMCD w/o Hypothesis 1 and 2 |  0.058 |
>
> > **Q2: How do these methods differ from data augmentation techniques used in other user modeling tasks?**
>
>
> A2: Sorry for the confusion. The differentiation of our CMCD approach from standard data augmentation techniques is primarily manifested in two key aspects:
>
> * Monotonicity Assumption: The Monotonicity Assumption stands as a fundamental theoretical cornerstone in the realm of Cognitive Diagnostics (CD). It specifically asserts that a student's proficiency demonstrates a monotonic correlation with the likelihood of providing a correct response to an exercise. Building upon this premise, we have introduced two data augmentation constraints, with experimental results validating the efficacy of our approach.
> * Theoretical Guarantees: We have provided theoretical assurances regarding the accuracy and convergence of the proposed data augmentation strategies. Detailed information on these theoretical guarantees can be found in section 5.2 of our work.
>
> In terms of practical application, we have also adapted some traditional data augmentation methods to the domain of cognitive diagnostics (specific baseline details can be found in A.3). Comparative analyses between these traditional methods and our CMCD approach have revealed superior performance of CMCD in both accuracy and fairness. Furthermore, our method demonstrates accelerated convergence rates (specific details can be found in section 6.2, RQ1, RQ2). We intend to emphasize these pivotal points in the revision.
>
> > **Q3: Is it possible to extend these methods to other user modeling tasks?**
>
> A3: Thank you for your thoughtful question. The core contribution of our paper is reflected in two aspects. Firstly, we conducted an analysis from a data perspective and identified that data sparsity could lead to unfairness and inaccuracy. We proposed achieving more accurate and fair cognitive diagnostics through the lens of data augmentation. Secondly, based on the unique monotonicity assumption in the field of cognitive diagnostics, we introduced two data augmentation constraints and provided theoretical guarantees.
>
> Regarding the first point, drawing from our past experiences, we observed that the data distributions in other user modeling tasks, such as recommendation systems, and the field of cognitive diagnostics generally exhibit similar trends. This suggests that the approach of addressing data sparsity to enhance accuracy and fairness in user modeling can be transferred. However, in tackling the challenge of data sparsity, our paper uniquely utilized the cognitive diagnostics domain's distinctive monotonicity assumption to augment data. By integrating this with the cognitive diagnostics paradigm, we provided corresponding theoretical guarantees. This method cannot be directly extended. In the future, we will explore how to leverage the specific characteristics of user modeling tasks to effectively address data sparsity issues.

---

> > ### Comment · Reviewer_ujU9 · 2024-08-08
> >
> > Thanks for the clarification which addresses most of my concerns. I will keep my score.

---

> > > ### Author Response · Authors · 2024-08-13
> > >
> > > Thanks for your prompt feedback and for acknowledging the value of our work! We will revise the paper in line with your suggestions. Once again, we appreciate your support and constructive feedback.

---

### Official Review · Reviewer_A4mA · 2024-07-12

**Soundness:** 4
**Presentation:** 3
**Contribution:** 3
**Rating:** 7
**Confidence:** 4

**Summary:**

This paper discusses the issues of fairness and accuracy in cognitive diagnostic tasks, which hold significant societal value and impact the fairness of education. Through an experimental perspective, the paper analyzes how data sparsity can lead to unfairness and inaccuracy in cognitive diagnostic tasks. It also leverages the specific assumption of monotonicity in the field of education to alleviate the challenges posed by data sparsity. Notably, the authors theoretically prove the convergence and efficacy of the proposed method. Finally, an analysis is conducted on real-world datasets, accompanied by thorough validation, providing substantial evidence for the efficacy of the approach.

**Strengths:**

- S1: This paper delves into a highly significant societal concern - the accuracy and fairness in cognitive diagnosis. To the best of my knowledge, this paper is the first to unveil how data sparsity can yield inaccuracies and unfairness in cognitive diagnosis.
- S2: From a data perspective, this paper effectively mitigates the challenge of data sparsity, steering clear of compromising the model's interpretability from a modeling standpoint. The motivation is sound, and the method is indeed compelling.
- S3: The paper undergoes validation from two key standpoints - experimental and theoretical. The validation methods are thorough and comprehensive, ensuring a robust evaluation.

**Weaknesses:**

- W1: I noticed that the experimental results at the end show an improvement in both the fairness and accuracy of diagnostics simultaneously, which seems to contradict the prevailing trade-off between fairness and accuracy. It would be beneficial for the paper to delve deeper into this issue, exploring whether this approach can be extended to other domains such as the field of recommender systems.
- W2: In the experimental section, I observed that the utility and fairness metrics are aligned. While I understand that fairness analysis is based on utility outcomes, this alignment could potentially mislead readers to some extent. Clarity on this relationship would enhance the understanding for the readers.

**Questions:**

See above weaknesses.

**Limitations:**

The authors have extensively discussed the limitations of their approach.

---

> ### Author Rebuttal · Authors · 2024-08-07
>
> > **W1: I noticed that the experimental results at the end show an improvement in both the fairness and accuracy of diagnostics simultaneously, which seems to contradict the prevailing trade-off between fairness and accuracy. It would be beneficial for the paper to delve deeper into this issue, exploring whether this approach can be extended to other domains such as the field of recommender systems.**
>
> Thank you for your constructive feedback. In fact, in the realm of fairness research, there does exist a trade-off between fairness and accuracy. This implies that while traditional fairness-aware methods can enhance fairness, they also tend to reduce utility. We have validated this phenomenon through experiments. Specifically, Table 1 displays the accuracy results of CMCD and the baselines, while Table 2 showcases the fairness results. Among these baselines, CD+Reg, CD+EO, and CD+DP are all considered fairness baselines. We observe that these baselines have to some extent improved the fairness of the original cognitive diagnostic model but have simultaneously decreased its performance.
>
> What sets our work apart in this article is our departure from traditional fairness approaches. We have approached the issue from a data perspective and, through extensive data analysis on real datasets, discovered that data sparsity can lead to inaccurate and unfair outcomes. The specific data analysis is presented in Section 4. This finding suggests that mitigating data sparsity in cognitive diagnostic tasks can alleviate the trade-off between fairness and utility. However, it is worth noting that in terms of the improvement in fairness, CMCD does not consistently outperform traditional fairness baselines and achieve state-of-the-art results. This discrepancy is due to our work considering both accuracy and fairness, highlighting the inherent trade-off between fairness and accuracy.
>
> Finally, we discuss the potential extension of our work to other domains, such as recommender systems. The data distribution in cognitive diagnostic fields and recommender systems exhibits similar trends. Therefore, alleviating data sparsity issues can potentially enhance both fairness and accuracy simultaneously, making the data-driven approach transferrable. However, our method relies on the unique monotonicity assumption in cognitive diagnostic fields to enhance data, combined with the cognitive diagnostic paradigm to provide corresponding theoretical guarantees. This approach cannot be directly extended. Thank you once again for your feedback. We will incorporate the relevant discussion in the revision.
>
>
> Table 1. The accuracy result of CMCD and baselines on NCDM （for the RMSE, MAE metric, the lower the better, for the AUC, ACC metric, the higher the better）
> | Baselines  | RMSE | MAE | AUC | ACC |
> | -------- | ------- | ------- | ------- | ------- |
> | origin  | 0.414 | 0.317 | 0.812 | 0.748 |
> | CD+Reg |  0.416 | 0.348 | 0.806 | 0.744 |
> | CD+EO |  0.420 | 0.321 | 0.803 | 0.740 |
> | CD+DP |  0.414 | 0.331 | 0.811 | 0.748 |
> | CMCD | **0.413** | **0.316** | **0.814** | **0.749** |
>
>
> Table 2. The fairness result of CMCD and baselines on NCDM (the lower, the better)
>
> | Baselines  | GF(RMSE) | GF(MAE) | GF(AUC) | GF(ACC) |
> | -------- | ------- | ------- | ------- | ------- |
> | origin  | 0.034 | 0.070 | 0.032 | 0.054 |
> | CD+Reg |  0.036 | 0.066 | 0.030 | 0.052 |
> | CD+EO |  0.040 | **0.054** | **0.027** | 0.057 |
> | CD+DP |  0.035 | 0.076 | 0.032 | **0.050** |
> | CMCD | **0.033** | 0.059 | 0.028 | **0.050** |
>
>
> > **W2: In the experimental section, I observed that the utility and fairness metrics are aligned. While I understand that fairness analysis is based on utility outcomes, this alignment could potentially mislead readers to some extent. Clarity on this relationship would enhance the understanding for the readers.**
>
> Sorry for the confusion. In this paper, the calculation of fairness metrics is based on user-oriented group fairness [1], which asserts that a fair model should deliver an equal level of utility performance across different user groups. Therefore, we express the performance gaps between the two groups accordingly. To simplify, we directly use performance metrics as a representation. To avoid any confusion, we will change our metric names in the revision. For example, in the fairness table, we will replace RMSE, MAE, AUC, ACC with GF(RMSE), GF(MAE), GF(AUC), GF(ACC).
>
> [1] Yunqi Li, Hanxiong Chen, Zuohui Fu, Yingqiang Ge, and Yongfeng Zhang. 2021. User-oriented
>  fairness in recommendation. In Proceedings of the Web Conference 2021. 624–632.

---

> > ### Comment · Reviewer_A4mA · 2024-08-08
> >
> > Thanks a lot for the authors' great effort. I have carefully read their response.
> > These responses satisfy my questions and reinforce my rating.

---

> > > ### Author Response · Authors · 2024-08-13
> > >
> > > We appreciate your recognition of the value of our work! We are pleased to have addressed your concerns and will revise the paper in accordance with your suggestions.

---

### Author Rebuttal · Authors · 2024-08-07

We sincerely appreciate all reviewers' time and efforts in reviewing our paper.  We extend our gratitude to each of them for offering constructive and valuable feedback, which we will use to enhance this work. Meanwhile, we are encouraged by the positive comments from reviewers, including:

* **Motivation:** "hold significant societal value and impact the fairness of education" (Reviewer A4mA), "a highly significant societal concern" (Reviewer A4mA), "a significant issue in the field of intelligent education"(Reviewer ujU9), "holds considerable research value" (Reviewer ujU9), "addressing the critical issue of data sparsity in cognitive diagnosis" (Reviewer xrHo)

* **Theoretical Contribution:** "the authors theoretically prove" (Reviewer A4mA), "The paper undergoes validation from two key standpoints - experimental and theoretical." (Reviewer A4mA), "provide theoretical guarantees" (Reviewer ujU9), "as demonstrated through theoretical analysis and experiments" (Reviewer xrHo)

* **Method:** "effectively mitigates the challenge of data sparsity" (Reviewer A4mA), " the method is indeed compelling" (Reviewer A4mA), "intuitive and rational" (Reviewer ujU9), " maintains interpretability while enhancing accuracy and fairness"(Reviewer xrHo)

* **Experimental Results:** "accompanied by thorough validation" (Reviewer A4mA), "The validation methods are thorough and comprehensive, ensuring a robust evaluation" (Reviewer A4mA), "released the corresponding code as open source and conducted extensive experiments"(Reviewer ujU9), "The use of real-world datasets and extensive experiments validates the efficacy of CMCD" (Reviewer xrHo), "providing a robust solution to a common problem" (Reviewer xrHo).

We have provided specific responses to each reviewer. We hope our responses can clarify all your confusion and alleviate all concerns. We thank all reviewers again. Looking forward to your reply!

---

### Decision · Program_Chairs · 2024-09-25

**Decision:**

Accept (poster)

**Comment:**

The submitted paper was reviewed by 3 knowledgeable reviewers all of whom recommended acceptance of the paper. The reviewers appreciate the importance of the considered problem for intelligent education, leveraging the monotonicity assumption for improved data augmentation, and the theoretical as well as extensive experimental evaluation demonstrating the benefits of the proposed approach. Thus, in line with the reviewers' unanimous recommendation, I recommend acceptance of the paper.
The authors are asked to prepare the camera-ready version by considering the reviewers' comments and in particular include the clarifications made during the discussion-phase as well as new results in the final paper.